# The Experiences and Views on Palliative Care of Older People with Multimorbidities, Their Family Caregivers and Professionals in a Spanish Hospital

**DOI:** 10.3390/healthcare10122489

**Published:** 2022-12-09

**Authors:** Laura Llop-Medina, Francisco Ródenas-Rigla, Jorge Garcés-Ferrer, Ascensión Doñate-Martínez

**Affiliations:** Polibienestar Research Institute, University of Valencia, 46022 Valencia, Spain

**Keywords:** chronic diseases, multimorbidity, qualitative research, older people, caregivers

## Abstract

The increasing prevalence of complex chronic diseases in the population over 65 years of age is causing a major impact on health systems. This study aims to explore the needs and preferences of the multimorbid patient and carers to improve the palliative care received. The perspective of professionals who work with this profile of patients was also taken into account. A qualitative study was conducted using semi-structured interviews with open-ended questions. Separate topic guides were developed for patients, careers and health professionals. We included 12 patients, 11 caregivers and 16 health professionals in Spain. The results showed multiple unmet needs of patients and families/caregivers, including feelings of uncertainty, a sense of fear, low awareness and knowledge about palliative care in non-malignant settings, and a desire to improve physical, psychosocial and financial status. A consistent lack of specialized psychosocial care for both patients and caregivers was expressed and professionals highlighted the need for holistic needs assessment and effective and early referral pathways to palliative care. There is a lack of institutional support for multimorbid older patients in need of palliative care and important barriers need to be addressed by health systems to face the significant increase in these patients.

## 1. Introduction

Population ageing is accelerating worldwide, increasing the incidence and prevalence of complex chronic diseases in the population over the age of 65 and, in turn, impacting health systems. Chronic diseases place a significant impact on older people’s quality of life, resulting not only in reduced physical functioning but also associated psychological distress and finance concerns [1]. People living with chronic diseases are also at a high risk of developing multimorbidity, understood as several long-term chronic diseases [2], leading to an increase in reliance on family and caregivers due to the progressive decline in health status. The impact of multimorbidity on functioning, quality of life and mortality risk is considerably greater than the sum of the individual effects of these diseases [3]. In addition, multimorbidity is also associated with higher rates of utilization of healthcare services and higher health costs [3]. The annual deaths due to patients with multimorbidities are projected to increase worldwide from 38 million in 2012 to 52 million by 2030 [4].

In the case of older patients with multimorbidities, the clinical uncertainty regarding their prognosis is very high [5]. Detecting the episodes of decompensations and unmet needs of these patients and their caregivers is essential for health systems to adequately attend to this profile of patients and avoid unnecessary complications or avoidable admissions to the emergency department. The experiences of people living with multimorbidities reveal multiple areas of need [6,7]. These needs can become complex and severe for end-stage patients, but the prognosis often remains uncertain for most non-cancer life-limiting conditions [8].

It Is widely known that the provision of palliative care has typically focused on cancer patients but its extension to patients with non-malignant diseases has been reported in the literature [9], such as among older patients with multimorbidities [9,10], which can significantly improve their quality of life and reduce the costs of the medical care they currently receive. There is increasing evidence showing that the provision of palliative care to non-cancer diseases improve patients’ symptoms burden and quality of life, resulting in a consistent pattern of reduced health-care use [11]. Although there is a difficulty in identifying the needs of older patients with multiple pathologies and the timing of referral to palliative care pathways due to the heterogeneity of this population and the co-occurrence of several diseases.

Solid evidence on early identification, needs assessment and mapped palliative care pathways to provide symptom management and the support to patients and their families is still lacking [12]. To improve the experience of multimorbid patients receiving palliative care and their families requires us to find out what their experience has been in the health system and the gaps or needs they have identified in their care. This qualitative study, therefore, sought to understand what constitutes successful specialist palliative care in the view of all stakeholders involved, older multimorbid patients with non-cancer diseases, their informal caregivers and the health professionals who carry out their work in the field of palliative care. The research question of this study was: “What are the experiences and views on palliative care of older people with multimorbidities, their family caregivers and professionals?”.

## 2. Materials and Methods

A qualitative research methodology was employed in order to explore participants’ perspectives and experiences regarding palliative care. One-to-one interviews with multimorbid older patients (aged 65 or over) and their families/carers (12 and 11, respectively), and two focus groups with health professionals (eight professionals in each focus group) who provided disease-directed services and palliative care in a Home hospitalization unit (HHU) in La Fe Hospital (Valencia, Spain) was conducted.

This approach allowed us to understand an in-depth perspective of palliative care involvement, the current status of palliative care services and how these can be improved. Ethics approval was granted by Ethics Committee on Research with Medicaments of Hospital La Fe in Spain (ref. 2019-013-1).

### 2.1. Identification of Participants

The Valencia-La Fe Health Department is one of the healthcare areas belonging to the Health System of the Autonomous Community of Valencia (Spain) and is recognized in Spain as a leading hospital in the management of various diseases, in the provision of home hospitalization services and in the search for the integration of all levels of care. La Fe Health Department has a case management plan for chronically ill patients, which cares for around 900 patients with complex conditions. This plan includes multidisciplinary teams between telemedicine and Hospital at Home Units, which coordinates care plans, home visits, scheduled phone calls and communication with primary care in order to keep chronic patients as stable as possible.

Purposive sampling was used to recruit patients, who were eligible if they were 65 years or older, diagnosed with multimorbidity and able to read and speak Spanish. Patients were excluded if they were living with or receiving cancer treatment or were cognitively impaired. Families/carers were eligible if they were cognitively intact and able to read and speak Spanish. Health professionals providing direct care for palliative care patients’ or managing palliative care services were targeted.

The participants were initially identified by a member of the clinical team of the HHU. Recruitment and interviews took place between November 2019 and March 2020. All of the individual interviews were conducted face-to-face at patient’s home by a researcher with experience in conducting qualitative research in previous studies. In the case of health professionals, the two focus groups took place in a meeting room of La Fe Hospital in Valencia and lasted one and a half and two hours, respectively.

### 2.2. Data Collection

Qualitative data were collected using semi-structured interviews and focus groups [13] using open-ended questions. Separate topic guides were developed for patients, families/carers and health professionals concerning disease symptoms and impact, experiences of services and care provided, management of exacerbations, needs for the future (for patients and families/carers) or palliative care pathways and integration of palliative care in the management of severe conditions (for health professionals). All of the interviews and focus groups were digitally recorded with consent and transcribed verbatim.

### 2.3. Data Analysis

The consolidated criteria for reporting qualitative studies (COREQ) checklist [14] was used to analyze the data which provides a systematic approach to sifting (Appendix A), charting and sorting data using the key themes and issues. Codes were given to segment the information and the key conclusions extracted from the interviews. Codes were compared across all interviews to identify similarities and differences to be further groups into categories, synthesizing and explaining large amounts of data. Connections and relationships between codes and categories were further explored enabling the development of themes and sub-themes. Atlas.ti software was used to analyze the data [15]. Two authors contributed to coding, developing frameworks and themes. We identified four main thematic categories and fifteen sub thematic categories presented in Table 1.

## 3. Results

Twenty participants were interviewed (12 patients, 11 caregivers) and sixteen health professionals participated in the focus groups. The mean age of the patients was 77.41 years (SD 7.82), there were eight men and four women, only three people had secondary or some university degree (six had primary education and three had none), most were married or living with a partner (nine) and all were retired. Five of them were in the palliative care patient program and the rest in chronic pluripathological complex case-management programme. The average age of the caregivers was 67 years (SD 8.54), all of them women (nine wives and two daughters), mostly retired (seven caregivers) or not working (three caregivers), who provided daily care (ten cases).

In the case of the professionals, the mean age was 47.42 years (SD 10.57), twelve women and four men, the majority nurses (nine cases) and physicians (five cases), although a social worker and a psychologist also participated. The majority worked in HHU (eight professionals), four worked in primary care and four in health management. The area of expertise was chronicity, palliative care and geriatrics in thirteen cases, primary care in two cases and medical management in one case. Moreover, although the majority would have previous training in palliative care (twelve cases), four professionals did not have such training.

### 3.1. Management of Symptoms

Interviews with the participants consistently reported the following symptoms by patients: breathlessness, lack of mobility, back pain, sadness, anxiety and respiratory failure. Patients (P) described the lack of mobility and fatigue as very traumatic, to similary with caregivers (C) faced many problems in daily care for patients due to lack of mobility of their patients. Health professionals (HP) highlighted the importance of having highly involved caregivers to keep patients’ symptoms under control, as most were receiving hospital care at home. Caregivers often felt overburdened and lacked information on how to deal with some of the symptoms of their relatives.

9P: “When I see that I don’t, that I cannot do things by myself, that others have to help me to wash up, to have a shower and so on… I feel sadness when I see that I do not do things as before. I don’t know how to tell you” (patient with multimorbidites, man).

9C: “I would like to have information, I mean the guidelines for example to made better movements to lift her, or to put her in bed” (caregiver, woman).

### 3.2. Health Service Experience

The patients and carers rated the care received at home by HHU very positively, highlighting positive aspects of this unit, such as the human quality of the professionals, the feeling of support from the healthcare team, and above all having the care of the hospital but with the patient at home. On the other hand, they highlighted the negative aspects of waiting times in Emergency Rooms (ER) and the lack of information regarding the patient’s condition while waiting in this service, as well as the absence of specialists to attend to emergencies. As for primary care, patients and carers remarked that there are excessive waiting lists.

Health professionals highlighted as fundamental to detect multimorbid patients and including them in the HHU programme which provides continuous telephone attention and home visits by physicians and nurses. The HHU identifies patients with multimorbidity who have palliative care needs and are included in the palliative management programme, then they work with the patient and family on the prevention of exacerbations, and explain how to detect an episode of decompensation, or when to notify or contact the nurse manager.

10C: “At hospital (ER) a different doctor visits you every time. You don’t have a monitoring with a single doctor… This is an inconvenient because they have to revise his/her history. Each one has his/her own way to see things not all of them say the same” (caregiver, woman).

12C: “The number of persons, when you arrange a doctor’s appointment… even if it is to request medications, you have to wait 5 or 6 days or a week” (caregiver, woman).

### 3.3. Views about Palliative Care

Patients and their families were not prepared to talk about palliative care and shied away from questions related to this topic. The use of the term palliative care has negative connotations for them, and they associated it with treatments for the last days of life and they reported not having much information about palliative care. No professionals had experienced palliative care during their training and they reported having their first contact with palliative care patients when they were already practising in their profession. There are no subjects during their training, and they remarked that their experience of palliative care depended on internal (health department) courses or training sessions.

9C: “What they say me now is that he is in palliative care, because maybe they need to do many medical tests, but it cannot solve anything. Since they are in charge of him he is getting better” (caregiver, woman).

1HP: “some course organized by the medical school years ago, to palliative care days like these, and mainly professional experience in the field” (MD specialist in Family and Community Medicine).

### 3.4. Palliative Care Needs

The interviews consistently reported that the main patients’ needs are highest level of mobility and independence, physiotherapy, personal support at home, public funding for dependent patients (assistive devices and personal support at home), psychological attention, availability of health provider’s time and drugs not funded by the system.

5P: “If I would lose mobility, my wife (current caregiver) she could not stand me up from the chair. In that case, yes, I would need to have a … a caregiver able to stand me up and so on” (patient with multimorbidities, man).

4P: “That someone comes to help me to stand up from here, to move on the bed in order to not disturb anybody. Because I am in that bed and I have to call to my wife or to this woman so they can turn me around in the bed” (patient with multimorbidities, man).

In terms of carers’ needs, interviewees reported that families or carers of patients need some support in areas such as loss of employment and income due to the need to reduce their working hours or even leave their jobs to care for their relative. In some cases, the loss of income was due to hiring support staff to care for their relative. Similarity, they also demanded financial support to meet the costs of long-term care.

1C: “Regarding dependence, if we were in a situation that we cannot dispose of these resources (savings), I do not know if public subsidies would cover or to what extent a person in my situation would go crazy because it would feel helpless” (caregiver, woman).

9C: “Now we also have to search privately other persons (hiring staff) that help the caregivers, because a single person cannot deal with this” (caregiver, woman).

Training to manage patients’ needs—such as guidelines, information or advice on how to deal with patients’ complex needs—was also highlighted by carers. Personalised care and more information from clinicians were needs highlighted by carers too.

7C: “We are facing a disease that even sometimes, both the patient and I we feel like a zero, I mean, blind, thus a more personalized attention, yes” (caregiver, woman).

9C: “I would like to have information, I mean the guidelines for example to made better movements to lift her, or to put her in bed” (caregiver, woman).

Carers reported the need to receive caregiving support. In some cases, caregiving involves 24 h a day and some carers expressed the need for substitute staff to allow them some respite, time and space for themselves in order to improve their own wellbeing and avoid feeling overburdened or burned out.

7C: “Yes, I would like to dispose of at least half an hour for myself alone to separate a bit from him, to distance myself and catch my breath, but it can’t be and, well, I adapt” (caregiver, woman).

7C: “I am not the first person I have told you this, being 24 h with a person around 2 consecutive years emotionally consumes you” (caregiver, woman).

The emotional overload was felt strongly and manifested in depressive symptoms, emotional fatigue, loneliness and burden. Caregivers expressed that they would like to receive emotional support to cope with these feelings.

10C: “I am very depressed, I am very depressed, and in fact I take one pill every day for that…His health status has declined and that sinks me because he does not treat me well, I mean, this is a vicious circle” (caregiver, woman).

In some cases, caregivers also expressed the need to have a physiotherapist for patients with limited mobility, and they considered that this would improve their quality of life and prevent ulcers or sores.

1C: “We decided to hire, privately, the services of a physiotherapist once per week. At the same time, this professional is subjected her to what he calls “passive physical exercise”, in arms and legs, which aims to alleviate the consequences of the patient’s sedentary lifestyle” (caregiver, woman).

Health professionals highlighted the lack of training in palliative care as a barrier to providing quality care to these patients, and the professionals interviewed expressed the need for a specific training pathway in palliative care to help them gain confidence to talk openly with patients and carers about palliative care. 

3HP: “when I was studying the degree they don’t teach us anything about palliative care, at least they didn’t teach me anything and I had to go on my own. As I am a tutor of residents because I had the opportunity in the month of formation that they give us to ask for a rotation with the HHU, and it was a real discovery of what Palliative Care was” (HHU doctor).

11HP: “there was only an elective subject that was called “palliative care”, although after having the experience of palliative care that I have, it was really something very abstract, very focused on the traditional model of palliative care” (MD specialist in Family and Community Medicine).

The ability to provide multidisciplinary care, addressing the psychosocial well-being of patients and carers was reported as limited by professionals. Time for personalised care addressing bereavement was highlighted as lacking.

4HP: “the health system is very medicalized, a lot, with a lot of nursing, but we lack physiotherapists, psychologists, social workers, that is very important for the patient and the family, little by little some have been included, but this is the future that should be sought” (HHU doctor).

10HP: “in primary care in terms of psychological, social and emotional support there is ... yes many shortcomings” (HP6 MD specialist in Family and Community Medicine).

Health professionals reported that there is a need for tools to identify older multimorbid patients in need of palliative care at an early stage and to systematise care protocols as is the case for oncology patients, so that the necessary resources can be planned for this patient profile.

9HP: “I believe that the cancer patient is more protected by specialized than the non-oncological patient” (nurse front line area HHU).

12HP: “We find patients at a very late stage and that is when you have to start looking for social resources, which will not arrive because they will die before, that if you had started to manage it months before because the patient could have been with more resources at home, better cared for, or you could have anticipated symptoms that you knew were going to appear and maybe the patient and the families were better informed and could detect decompensation earlier, but we treated patients in the acute processes” (social worker in HHU).

Finally, health professionals considered the scarce support and funding for patients and also for family carers to be problematic, as this can cause the family to give up leading to a very problematic scenario with hospitalised patients at home.

7HP: “The aid is usually paid later, the family has to spend it first” (nurse primary care).

1HP: “You can have very good pain control, very good control of other symptoms, and have a patient who is bedridden or who cannot go out, or who cannot enjoy their quality of life because they lack other resources that you cannot give them, and that is very important” (MD specialist in Family and Community Medicine).

We will summarize the main barriers and facilitators for the integration of palliative care in Table 2.

## 4. Discussion

There is increasing evidence that the implementation of palliative care generates clinically significant improvements in the patient’s quality of life [11]. The findings of this study suggest that there are unmet needs of older patients with multimorbidities and their families/carers for timely palliative care, including feeling uncertain and a sense of fear, poor awareness and knowledge on palliative care in non-malignant settings and the desire for improved physical, psychosocial and financial status as suggested in previous studies [12,16,17]. It would be interesting to relate these needs to factors such as frailty in these patients. In the Spanish health context, the correlation between frailty and end-of-life illness trajectories and survival has been demonstrated [18]. Caregivers also reported a high level of frustration and anxiety in this study, not only associated with their own health conditions but also with making care decisions without adequate support.

A constant lack of specialized psychosocial care for both patients and caregivers was manifested, which generates unmet needs, as seen in other studies [19,20]. Hashemi reported the psychosocial needs of family caregivers of cancer patients highlighted the importance of meeting the needs of family members to improve patients’ quality of life [19]. McIlfatrick underlined the need for supportive care for caregivers of people with heart failure at the end of life, including the need for emotional support, information about prognosis and advice on where and how to access the necessary support for themselves [20]. This study also highlights the lack of psychosocial support for older patients with multimorbidities and their caregivers.

Unmet needs for health professionals and the healthcare system were also identified including the lack of coherent support for patients and families, early identification, holistic needs assessment and effective referral pathways for palliative care input. The need expressed by professionals to have highly involved caregivers contrasts with the testimonies of caregivers who often feel overloaded and burned by the intensity of care they must provide, often 24 h a day, in patients with a high dependence and burden of symptoms. Responses from health professionals consistently lead to the consideration of shifting palliative care systems towards a holistic and anticipatory planning approach, this is consistent with recent studies with other patient profiles, people with dementia [21] or heart failure [22].

Understanding and addressing palliative care needs may support the development, implementation, mechanisms, and evaluation of the integration of palliative care for older people with advanced diseases, for both research and practice. While palliative care has been widely applied to cancer patients to improve their quality of life [23,24], it is currently not widely applied to patients with non-malignant diseases, although there is already evidence that referring these patients to palliative care pathways decreases unnecessary hospitalizations and emergency department visits [25]. However, uncertainty about the prognosis of older multimorbid patients is a barrier to referral to palliative care, and the surprise question (“Would I be surprised if this patient died next year?”) has been used [26] to try to recognize patients with non-malignant diseases who may need palliative care, although it has not proved to be an accurate predictor according to the Downar’s study [27]. In the Spanish context, Gómez-Batiste et al. [28] propose a 2-year mortality prognostic approach for patients with advanced chronic conditions based on the palliative care need (PCN) items of the Palliative Needs (NECPAL).

Additionally, the development of evidence-based tools for early identification and needs assessment with optimized clinical pathways remains a priority. Recent studies [29] have presented mathematical models that can predict frailty and mortality in older patients and support the identification of older patients in need of palliative care.

In addition, specific training in palliative care should be encouraged for all health professionals, with emphasis on the different patient profiles that may be targeted. In particular, the profile of older patients with complex chronic diseases who, although they have an uncertain prognosis due to their multiple pathologies, can clearly benefit from early referral to palliative care. Likewise, palliative care should be organized at institutional level in multidisciplinary teams that holistically address the needs of patients and their families with special attention to psychosocial well-being and including financial benefits to meet their long-term care needs.

## 5. Conclusions

This study highlights very significant issues both for our societies in general and for the palliative care sector in particular: care for the older people affected by chronic illnesses, the burden of carers, the scarce development of services that should attend to these emerging needs, even in home care, which is still very recent in the Valencian context as, currently, only a few hospitals have a home care service. Research into palliative care in this area, especially using qualitative techniques, is essential to assess needs, limits and possible solutions. The needs and shortcomings of palliative care for chronically older patients reported by health professionals can guide the development of future public health policies. It is essential that palliative care includes pathways for older patients with non-malignant diseases, as this is one of the main challenges our societies will face in the future.

## 6. Strengths and Limitations

This study provided a comprehensive view of palliative care needs in older patients with multimorbidities. The purposive sampling allowed us to cover the different profiles of multimorbid patients attended by the HHU of the hospital where the study was carried out, as well as the professionals who work with this profile of patients, obtaining representativeness of all of them. Although the sampling was carefully undertaken, its representativeness is limited given that it was a purposive sampling managed by the clinical team of the UHD, it is worth mentioning the possible selection bias and that the participants who chose to participate may have had a more satisfactory experience than those who declined to participate. All study participants recruited were white Spanish, which may not reflect the experiences of those from minority ethnic communities, and future work is needed to capture their views. The study was carried out in a single hospital and the results cannot be extrapolated to other hospitals.

Our findings suggested the need of institutional support for multimorbid older patients and identified important barriers that need to be addressed by healthcare systems facing a significant increase in these patients. Although the study was conducted before healthcare systems were severely threatened by the SARS-CoV-2 pandemic, the findings remain relevant to the profile of multimorbid older patients. However, post-pandemic changes should be taken into account in future studies, such as those identified by Franchini [30].

## Figures and Tables

**Table 1 healthcare-10-02489-t001:** Main and sub-thematic categories.

Main Thematic Categories	Sub-Thematic Categories
Management of symptoms	Health status, aggravation of symptoms, caregiver overburden.
Health service experience	Access to support, views about Home Hospitalization Unit, detection of multimorbid patients.
Views about palliative care	Negative connotation of palliative care term, lack of professional’s training in palliative care.
Palliative care needs	Physical needs, emotional needs, spiritual needs, social needs, economical needs, caregiver specific needs, professionals specific needs

**Table 2 healthcare-10-02489-t002:** Perceived barriers and facilitators for Palliative care integration.

	Barriers	Facilitators
Patients	Lack of knowledge concerning what palliative care is. Excessive waiting lists in some services. Waiting time in ER. Lack of financial resources.	Home based services. Avoid invasive treatments. Social resources and assistive devices. Physiotherapy. Availability of 24-h medical care by telephone.
Caregivers	Lack of knowledge concerning what palliative care is. Lack of information in waiting rooms. Lack of emotional support. Lack of time with the healthcare team. Lack of financial resources	Being supported for HP. More information by healthcare team. Receiving guidelines on how to care for palliative patients. Financial support. Time and space for their wellbeing. Availability of 24-h medical care by telephone.
Healthcare professionals	Lack of confidence in talking about palliative care (with patients and caregivers). Lack of training in palliative care. Lack of time for communicate with carers. Inexistence of early referrals to palliative care for this patient profile. Palliative care usually applied to last days of life. Lack of financial support for patients.	Tools to identify earlier palliative care patients (older people with multimorbidities). Time to work with patients and their caregivers. Support on how to deal with sensitive issues. More and specialised resources for chronicity.

## Data Availability

Not applicable.

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
