# Peer review of "The Experiences and Views on Palliative Care of Older People with Multimorbidities, Their Family Caregivers and Professionals in a Spanish Hospital"

_healthcare, 2022, doi:10.3390/healthcare10122489_

Round 1

Reviewer 1 Report

It isn't that this is a poor-quality article. The topic is important, and the authors write clearly. But: did they discover anything that hasn't been known and said many times before? The burdens of home-care on the spouses of very sick elderly persons have been documented hundreds of times. The lack of social support, limitations of what professionals can provide, and the "medicalization" of palliative care, seem to be part of the scene in developed countries. The patients in this study have the benefit of home care, a health care system that appears to be functioning and orderly- and yet there are all the unmet needs documented in this study. They are, one might say, the relatively lucky ones. They have family members to care for them. They have lived long lives. They may not have adequate support or information, but they have not been abandoned by the system or the society. Will better palliative care solve all their problems? It is useless to become nostalgic for some traditional pattern of huge multi-generational families, where the tasks of home care could be spread around. Or where even bed-ridden very sick persons had some particular roles in family life that made them feel valued.  The situations described in this essay are the conditions we now have.

Author Response

Response to Reviewer 1 Comments

Point 1: It isn't that this is a poor-quality article. The topic is important, and the authors write clearly.

Response 1: Dear reviewer, thank you very much for your work and for your comments which have undoubtedly helped to improve the quality of our work.

Point 2: But: did they discover anything that hasn't been known and said many times before? The burdens of home-care on the spouses of very sick elderly persons have been documented hundreds of times. The lack of social support, limitations of what professionals can provide, and the "medicalization" of palliative care, seem to be part of the scene in developed countries. The patients in this study have the benefit of home care, a health care system that appears to be functioning and orderly- and yet there are all the unmet needs documented in this study. They are, one might say, the relatively lucky ones. They have family members to care for them. They have lived long lives. They may not have adequate support or information, but they have not been abandoned by the system or the society. Will better palliative care solve all their problems? It is useless to become nostalgic for some traditional pattern of huge multi-generational families, where the tasks of home care could be spread around. Or where even bed-ridden very sick persons had some particular roles in family life that made them feel valued.  The situations described in this essay are the conditions we now have.

Response 2: We agree with the reviewer that we are describing the current situation, but this description should serve to move forward, to show that we still have many problems to solve. Qualitative methods can be very useful in that process. We also agree that we cannot go back to previous situations in which the family assumes all the responsibility; we believe in the need to improve public health and social care systems to relieve families and help them.

Our article aims to highlight very significant issues both for our societies in general and for the palliative care sector in particular: the care of the older people affected by chronic illness, the burden of carers, the underdevelopment of services that should address these emerging needs, even in home care, which is still very recent in the Valencian society as a few hospitals have a home care service right now. All in all, our study provides an insight into the current state of this care and in no way suggests that the solution to the problems detected will be found by returning to the traditional system of large multigenerational families, but rather that the Valencian healthcare system should improve these services in order to provide a solution to some of these problems. Furthermore, we believe that research into palliative care in this area, especially using qualitative techniques, is essential to assess needs, limits and possible solutions. We also consider that our study has also gathered the point of view of professionals who can guide the development of future public health policies and it is interesting that palliative care should include pathways for chronic older patients with non-malignant diseases, as this is one of the major problems that our societies will face in the future.

Reviewer 2 Report

I attach my comments and suggestions

Author Response

Response to Reviewer 2 Comments

Point 1: This is an interesting paper and brings forward some very good findings.

Response 1. Dear reviewer, thank you very much for your work and for your comments which have undoubtedly helped to improve the quality of our work.

Point 2: However, the English is often confusing and there is a need for many changes: Line 11 to Line 340.

Response 2: All suggested changes have been implemented.

Point 3: There is a lack of clarity in the methods used. In the Methods section the paper talks of Focus

Groups with professionals but then interviews are discussed in the results. This needs to be clearer and details of the Focus groups should be given how many, how long.

Response 3: The methodology section has been rewritten, providing clarity on the interviews (with patients and carers) and focus groups carried out with professionals. The section has been rewritten as follows:

Materials and Methods

A qualitative research methodology was employed in order to explore participants’ perspectives and experiences regarding Palliative Care. One-to-one interviews with multimorbid older patients (aged 65 or over) and their families/carers (12 and 11, respectively) were performed. Also, two focus groups with health professionals (8 professionals in each focus group) who provided disease-directed services and Palliative Care in a Home hospitalization unit (HHU) in La Fe Hospital (Spain) were conducted.

This approach allowed us to understand an in-depth perspective of Palliative Care involvement, the current status of Palliative Care services and how these can be improved. Ethics approval was granted by the Ethics Committee on Research with Medicaments of Hospital La Fe in Spain (ref. 2019-013-1).

2.1. Identification of participants

Purposeful sampling was used to recruit the patient’s ant they were eligible if they were aged 65 or over, diagnosed with multimorbidities and were able to read and speak in Spanish. Patients were excluded if they lived or received treatment for cancer or were cognitively impaired. Families/carers were eligible if they were cognitively intact and were able to read and speak Spanish. Health professionals providing direct care for Palliative Care patients or managing Palliative Care services were targeted.

The participants were initially identified by a member of the clinical team of the HHU. Recruitment and interviews took place between November 2019 and March 2020. All the individual interviews were conducted face-to-face at the patient’s home by a researcher with experience in conducting qualitative research in previous studies (LLL). In the case of health professionals, the two focus groups took place in a meeting room of La Fe Hospital in Valencia and lasted one and a half and two hours respectively. 

2.2 Data collection

Qualitative data were collected using semi-structured interviews and focus groups [13] using open-ended questions. Separate topic guides were developed for patients, families/carers and health professionals concerning disease symptoms and impact, experiences of services and care provided, management of exacerbations, needs for the future (for patients and families/carers) or Palliative Care pathways and integration of Palliative Care in the management of severe conditions (for health professionals). All the interviews and focus groups were digitally recorded with consent and transcribed verbatim

Point 4: It would be advisable to have the revised manuscript checked by an English speaker before submission.

Response 4: The manuscript has been reviewed by an English native speaker.

Reviewer 3 Report

The paper highlights extremely significant issues both for our societies in general and for palliative care sector in particular: the caring of the older affected by chronic diseases, the burden of the caregivers, the underdevelopment of the services which should tackle these emerging needs. Palliative care research in the field, especially through qualitative techniques, is essential to assess needs, limits and possible solutions. Authors performed the study consistently in terms of qualitative methodology. The paper is also well written, and I could not find any insurmountable limits. Nonetheless, I found some deficiencies, easily manageable by authors.

-          The research question only partially fits the title of the paper. The research should in fact answer to the question: "What are the experiences and views on palliative care of older people with multimorbidities, their family caregivers and professionals?”. Readers would expect to find, since the title of the article, some insight about representations of palliative care organization/professionals/concept. I suggest to simply modify the title or better explain how to read the research question.

-          Since authors chose purposive sampling of the patients/caregivers, initially managed directly by a member of the clinical team of the HHU, and there is likelihood that patients/caregivers who choose to adhere had good experience with the services, selection biases should be more stressed in the proper section. Similarly, since there is no comparison with other services (La Fe hospital), the resulting limitations should be better highlighted in the paper.

-          Considering study was conducted in one hospital, and especially in one unit such HHU, more specifics in terms of quantitative and qualitative data about it would be useful. Also, I comprehended only partially the perspective authors chose (Grounded Theory or other framework).

-          Authors during selection and analysis/interpretation process on data gathered from interviews and FG have considered previous experiences patients and caregivers enrolled may have had, such palliative care /services, as patient exposed to individual consultation and/or as proxy, for instance as caregiver?

-          Since authors adopted qualitative, the context (in this case, from Spanish society to valencian university hospital) is crucial. Therefore, I suggest authors may cite more situated studies. Spanish authors published many papers in the last years about needs experienced by patients with chronic complex conditions and their family. Similarly, authors may find different papers on these topics from other Mediterranean countries, such Italy. Similarly, early identification and correlation between frailty, palliative care, death and needs are topics widely studied by Spanish author like Amblas, Gómez-Batiste and others (For instance: Gómez-Batiste X, Turrillas P, Tebé C, Calsina-Berna A, Amblàs-Novellas J. NECPAL tool prognostication in advanced chronic illness: a rapid review and expert consensus. BMJ Support Palliat Care. 2022 May;12(e1):e10-e20. doi: 10.1136/bmjspcare-2019-002126. Epub 2020 Apr 2. PMID: 32241958; Amblàs-Novellas J, Murray SA, Oller R, Torné A, Martori JC, Moine S, Latorre-Vallbona N, Espaulella J, Santaeugènia SJ, Gómez-Batiste X. Frailty degree and illness trajectories in older people towards the end-of-life: a prospective observational study. BMJ Open. 2021 Apr 21;11(4):e042645. doi: 10.1136/bmjopen-2020-042645. PMID: 33883149; PMCID: PMC8061834.)

Author Response

Response to Reviewer 3 Comments

Point 1. The paper highlights extremely significant issues both for our societies in general and for palliative care sector in particular: the caring of the older affected by chronic diseases, the burden of the caregivers, the underdevelopment of the services which should tackle these emerging needs. Palliative care research in the field, especially through qualitative techniques, is essential to assess needs, limits and possible solutions. Authors performed the study consistently in terms of qualitative methodology. The paper is also well written, and I could not find any insurmountable limits.

Response 1: Dear reviewer, thank you very much for your work and for your comments which have undoubtedly helped to improve the quality of our work.

Point 2: The research question only partially fits the title of the paper. The research should in fact answer to the question: "What are the experiences and views on palliative care of older people with multimorbidities, their family caregivers and professionals?”. Readers would expect to find, since the title of the article, some insight about representations of palliative care organization/professionals/concept. I suggest to simply modify the title or better explain how to read the research question.

Response 2: As suggested, the title and research question of the manuscript have been changed. Thanks for your suggestion and feedback on that.

Point 3: Since authors chose purposive sampling of the patients/caregivers, initially managed directly by a member of the clinical team of the HHU, and there is likelihood that patients/caregivers who choose to adhere had good experience with the services, selection biases should be more stressed in the proper section. Similarly, since there is no comparison with other services (La Fe hospital), the resulting limitations should be better highlighted in the paper

Response 3: Thanks for highlighting this point. Following your suggestions, the section on strengths and limitations has been modified to read as follows:

“This study provided a comprehensive view of Palliative Care needs in older patients with multimorbidities. The purposive sampling allowed us to cover the different profiles of multimorbid patients attended by the HHU of the hospital where the study was carried out, as well as the professionals who work with this profile of patients, obtaining representativeness of all of them. Although the sampling was carefully undertaken, its representativeness is limited given that  patients were identified and invited following a purposive sampling that was managed by the clinical team of the HHU. Thus, it is worth mentioning the possible selection bias and that the participants who chose to participate may had a more satisfactory experience than those who declined to participate. All study participants recruited were white Spanish which may not reflect the experiences of those from minority ethnic communities, and future work is needed to capture their views. The study was carried out in a single hospital and the results cannot be extrapolated to other hospitals.

Our findings suggested the need of institutional support for multimorbid older patients and identified important barriers that need to be addressed by healthcare systems facing a significant increase in these patients. Although the study was conducted before healthcare systems were severely threatened by the SARS-COV2 pandemic, the findings remain relevant to the profile of multimorbid older patients. However, post-pandemic changes should be taken into account in future studies, such as those identified by Franchini [28].”

Point 4: Considering study was conducted in one hospital, and especially in one unit such HHU, more specifics in terms of quantitative and qualitative data about it would be useful. Also, I comprehended only partially the perspective authors chose (Grounded Theory or other framework).

Response 4: The following paragraph has been added to the section on materials and methods, identification of participants of participants:

“The Valencia-La Fe Health Department is one of the health care areas belonging to the Health System of the Autonomous Community of Valencia (Spain) and is recognized in Spain as a leading hospital in the management of various diseases, in the provision of home hospitalization services and in the search for the integration of all levels of care. This hospital covers a geographical area with a population of around 300,000 inhabitants. La Fe Health Department has a case management plan for chronically ill patients, which cares for around 900 patients with complex conditions. This plan includes multidisciplinary teams between telemedicine and Hospital at Home Units, which coordinates care plans, home visits, scheduled phone calls and communication with primary care in order to keep chronic patients as stable as possible.

The study employed framework analysis, an inherently comparative form of thematic analysis that employs an organised structure of inductively and deductively derived themes (i.e. a framework) to conduct a cross-sectional analysis through a combination of description and abstraction of data. This information can be found in the supplementary material 1

Point 5: Authors during selection and analysis/interpretation process on data gathered from interviews and FG have considered previous experiences patients and caregivers enrolled may have had, such palliative care /services, as patient exposed to individual consultation and/or as proxy, for instance as caregiver?

Response 5: The previous experience of patients and caregivers is reflected in their opinions collected in the analysis and interpretation of the results; but they were not considered as criteria in the selection process.

Point 6:  Since authors adopted qualitative, the context (in this case, from Spanish society to valencian university hospital) is crucial. Therefore, I suggest authors may cite more situated studies. Spanish authors published many papers in the last years about needs experienced by patients with chronic complex conditions and their family. Similarly, authors may find different papers on these topics from other Mediterranean countries, such Italy. Similarly, early identification and correlation between frailty, palliative care, death and needs are topics widely studied by Spanish author like Amblas, Gómez-Batiste and others (For instance: Gómez-Batiste X, Turrillas P, Tebé C, Calsina-Berna A, Amblàs-Novellas J. NECPAL tool prognostication in advanced chronic illness: a rapid review and expert consensus. BMJ Support Palliat Care. 2022 May;12(e1):e10-e20. doi: 10.1136/bmjspcare-2019-002126. Epub 2020 Apr 2. PMID: 32241958; Amblàs-Novellas J, Murray SA, Oller R, Torné A, Martori JC, Moine S, Latorre-Vallbona N, Espaulella J, Santaeugènia SJ, Gómez-Batiste X. Frailty degree and illness trajectories in older people towards the end-of-life: a prospective observational study. BMJ Open. 2021 Apr 21;11(4):e042645. doi: 10.1136/bmjopen-2020-042645. PMID: 33883149; PMCID: PMC8061834.)

Response 6: Thank you so much for this suggestion, which has been addressed including the mentioned references within the manuscript. Additionally, the text gathers now more context of the healthcare in the Autonomous Community of Valencia in order to provide a deeper framework of our study.
